# Protein Kinase A Regulates Autophagy-Associated Proteins Impacting Growth and Virulence of *Aspergillus fumigatus*

**DOI:** 10.3390/jof8040354

**Published:** 2022-03-30

**Authors:** E. Keats Shwab, Praveen R. Juvvadi, Shareef K. Shaheen, John Allen, Greg Waitt, Erik J. Soderblom, Yohannes G. Asfaw, M. Arthur Moseley, William J. Steinbach

**Affiliations:** 1Department of Pediatrics, Division of Pediatric Infectious Diseases, Duke University Medical Center, Durham, NC 27710, USA; keats.shwab@duke.edu (E.K.S.); praveen.juvvadi@duke.edu (P.R.J.); shareef.k.shaheen@gmail.com (S.K.S.); john.allen427@duke.edu (J.A.IV); 2Duke Proteomics Core Facility, Institute for Genome Sciences and Policy, Duke University, Durham, NC 27701, USA; greg.waitt@duke.edu (G.W.); erik.soderblom@duke.edu (E.J.S.); arthur.moseley@duke.edu (M.A.M.); 3Department of Laboratory Animal Resources, Duke University Medical Center, Durham, NC 27710, USA; yohannes.asfaw@duke.edu; 4Department of Molecular Genetics & Microbiology, Duke University Medical Center, Durham, NC 27710, USA

**Keywords:** *Aspergillus*, aspergillosis, protein kinase A, autophagy, nutrient sensing, phosphorylation, pathogenesis, cell wall, filamentous fungi, proteomics

## Abstract

Cellular recycling via autophagy-associated proteins is a key catabolic pathway critical to invasive fungal pathogen growth and virulence in the nutrient-limited host environment. Protein kinase A (PKA) is vital for the growth and virulence of numerous fungal pathogens. However, the underlying basis for its regulation of pathogenesis remains poorly understood in any species. Our *Aspergillus fumigatus* PKA-dependent whole proteome and phosphoproteome studies employing advanced mass spectroscopic approaches identified numerous previously undefined PKA-regulated proteins in catabolic pathways. Here, we demonstrate reciprocal inhibition of autophagy and PKA activity, and identify 16 autophagy-associated proteins as likely novel PKA-regulated effectors. We characterize the novel PKA-phosphoregulated sorting nexin Atg20, and demonstrate its importance for growth, cell wall stress response, and virulence of *A. fumigatus* in a murine infection model. Additionally, we identify physical and functional interaction of Atg20 with previously characterized sorting nexin Atg24. Furthermore, we demonstrate the importance of additional uncharacterized PKA-regulated putative autophagy-associated proteins to hyphal growth. Our data presented here indicate that PKA regulates the autophagy pathway much more extensively than previously known, including targeting of novel effector proteins with fungal-specific functions important for invasive disease.

## 1. Introduction

Protein kinase A (PKA) is a master regulator of eukaryotic metabolism, and essential for the virulence of numerous pathogens, including the fungus *Aspergillus fumigatus,* a leading infectious killer known to cause invasive aspergillosis (IA) in immunocompromised patients [1,2,3,4,5]. However, much remains unknown regarding its regulatory circuitry over critical cellular metabolic pathways, as only a few direct PKA effectors have been characterized in any species [6,7,8,9,10]. Our recent comprehensive analysis of the *A. fumigatus* PKA whole proteome and phosphoproteome enabled us to obtain the broadest in-depth proteome coverage of both direct and indirect PKA target proteins to date [11]. This work identified autophagic catabolism as a major functional category regulated by PKA that has not yet been characterized. Autophagy has previously been linked to fungal pathogenesis in the basidiomycete yeast *Cryptococcus neoformans,* wherein autophagy proteins Atg1, Atg7, Atg8, Atg9, and Vps34 each contribute to virulence [12,13]. Interestingly, even though all of these proteins are required for autophagy in *C. neoformans*, each impacted virulence to a different degree, suggesting that the influence of autophagy-associated proteins on pathogenesis may stem from potentially fungal-specific functions distinct from core autophagy processes. However, PKA regulation of fungal autophagy remains largely undescribed. Previously, only in the yeast *Saccharomyces cerevisiae* was PKA shown to regulate autophagy via phosphorylation of the autophagosome-associated Atg1–Atg13 protein complex [14,15]. However, among filamentous fungal pathogens, information on PKA regulation of autophagy is very limited. Only in the ascomycete plant fungal pathogen *Fusarium graminearum* was the PKA regulatory subunit mutant found to display defects in autophagy [16]. These limited studies illustrate the importance of autophagy pathways for fungal growth and disease, yet underscore the need to fully characterize PKA effectors in these pathways and their regulation in invasive fungal disease.

We recently employed high resolution and quantitative liquid chromatography-tandem mass spectroscopy (LC-MS/MS) to examine the global effects of PKA on both the total proteome and the phosphoproteome of *A. fumigatus* [11]. This study employed a robust methodology in which global effects of PKA on total protein and phosphoprotein levels across the *A. fumigatus* whole proteome were compared between the *A. fumigatus* wild-type (WT) and primary PKA catalytic subunit deletion strain (Δ*pkaC1*). Enhanced whole proteome and phosphoproteome coverage was achieved by proteolytic digestion with multiple proteases and complementary dual orthogonal approaches including TiO_2_-phosphopeptide and PKA-target motif specific antibodies enrichments to provide broader and in-depth coverage into phosphosite identification. Using this strategy, we identified autophagic catabolism as a primary functional category of proteins enriched in datasets for both altered protein expression and altered phosphorylation. This led to our characterization of two novel direct PKA target proteins, Atg24 and Not4, associated with catabolic recycling pathways, using targeted mutagenesis, molecular modeling, and IA murine models which demonstrated their requirement for *A. fumigatus* virulence. Furthermore, we found that PKA-dependent phosphorylation of specific target sites modulated their function, and also the response to antifungals targeting the cell wall and translation machinery.

Despite this progress, a comprehensive understanding of PKA regulation of autophagic pathways is lacking in any organism. Defining PKA-dependent regulation of our newly identified autophagy effector proteins during metabolic adaptation to the nutrient-limited host environment is necessary for understanding the molecular mechanisms of *A. fumigatus* pathogenesis, and future exploitation of fungal-specific pathways to combat invasive aspergillosis. In this study, we further investigate PKA regulation of autophagy in *A. fumigatus* by describing important roles for several identified novel and fungal-specific PKA-target autophagy-associated proteins in fungal growth and pathogenesis.

## 2. Materials and Methods

### 2.1. Quantitative LC-MS/MS Analysis

Protein from each sample was extracted via bead-homogenization with 8 M Urea. Either LysC or GluC was added and allowed to proceed for 18 h. After digestion, protein was subjected to antibody enrichment, trypsin digestion, and finally, TiO_2_ enrichment, as applicable. Phosphopeptide enrichments used TiO_2_ spin tips (GL Sciences, Torrance, CA, USA). For anti-PKA phosphopeptide antibody enrichment, resuspended peptides were then transferred in immunoaffinity purification buffer directly onto pre-aliquoted phospho-PKA substrate beads (Cell Signaling Technology, Danvers, MA, USA). Immunoprecipitation was performed for 2 h at 4 °C. Combined eluents were further enriched using TiO_2_ spin tips. Quantitative LC-MS/MS was performed using a nanoAcquity UPLC system (Waters Corp., Milford, MA, USA) coupled to a Thermo Fusion Lumos high-resolution accurate mass tandem mass spectrometer via a nanoelectrospray ionization source. Analytical separation was performed using a 1.7 μm Acquity BEH130 C18 75 μm × 250 mm column (Waters Corp.) with a 90 min linear gradient of 3 to 30% acetonitrile (Open analysis, 5 to 30% acetonitrile). Data collection on the Fusion Lumos mass spectrometer was performed in a data-dependent acquisition (DDA) mode of acquisition. Following 9 total UPLC-MS/MS analyses for each open analysis set, data were imported into Proteome Discoverer 2.2 (Thermo Scientific Inc., Waltham, MA, USA). Relative peptide abundance was calculated based on area-under-the-curve of the selected ion chromatograms. The MS/MS data were searched against the NCBI_*Apergillus fumigatus* database. Mascot Distiller and Mascot Server (v 2.5, Matrix Science, Boston, MA, USA) were utilized to produce fragment ion spectra, and to perform the database searches. To identify differences between WT and Δ*pkaC1* groups, a two-tailed *t*-test was performed on log_2_ protein or phosphopeptide intensities. Fold changes were calculated by ratioing the average intensity. To consider something differentially expressed between the groups, a *p*-value < 0.05 and a fold-change ≥ 2 were required.

### 2.2. Construction of Gene Deletions and Mutations in A. fumigatus

Mutant strains were generated as described previously [17]. Briefly, GFP-labeling of proteins was accomplished by insertion of the appropriate coding region 5′ to, and in frame with, the *egfp* coding region in a plasmid vector, followed by transformation of the *akuB^KU80^* strain of *A. fumigatus*, and screening via hygromycin B or *pyrG* selection as described [18,19]. RFP-labeling of Atg8 was accomplished by insertion of the *atg8* coding sequence and ~1 kb of downstream genomic flanking terminator sequence 3′ to the RFP coding sequence without stop codon in a plasmid, as well as insertion of a ~1 kb upstream flanking native promoter region of *atg8* 5′ to the RFP sequence, a hygromycin B resistance cassette 3′ to the *atg8* coding and terminator sequences, and ~1 kb of downstream flanking genomic sequence 3′ to the hygromycin B cassette to facilitate homologous recombination. This construct was then used to transform wild-type and Δ*pkaC1 A. fumigatus* strains, followed by hygromycin B selection. Gene deletions were accomplished via replacement of the open reading frame of interest with the *Aspergillus parasiticus pyrG* selectable marker. Deletions were verified via PCR and Southern blotting as described previously [20]. Plasmid constructs for site-directed mutagenesis were generated via fusion PCR as described previously [17]. All constructs and mutant strains generated were verified via DNA sequencing.

### 2.3. Assessment of Radial Growth, Conidiation, and Antifungal Compound Susceptibility

For radial growth assays, conidia (10^4^) were point inoculated on agar media in petri plates in triplicate, and incubated for 4 or 5 days at 37 °C. Media used included standard glucose minimal medium (GMM) (1% glucose, pH 6.5), as well as modified nitrogen-free GMM lacking NaNO_3_, or glucose-free minimal media to induce carbon and nitrogen starvation conditions, respectively, and GMM containing 0.5 µg/mL or 4 µg/mL caspofungin (CSP) to induce cell wall stress conditions. The mean radial growth rates for each of the strains were compared statistically by Student’s *t*-test.

### 2.4. Protein Extraction and Purification

*A. fumigatus* strains were cultured in liquid GMM with 250 rpm shaking for 24 h at 37 °C. For carbon and nitrogen starvation, strains were cultured in GMM for 20 h, then mycelia were filtered and transferred to carbon and nitrogen-free medium for an additional 4 h with shaking at 37 °C. Total cell lysate was obtained by homogenizing mycelia (1 g wet weight) using a mortar and pestle as previously described [17,18,21]. Total protein in the crude extracts was quantified by the Bradford method, and samples were normalized to contain 10 mg protein each. GFP-Trap^®^ (ChromoTek, Planegg-Martinsried, Germany) affinity purifications were performed from crude extracts according to the manufacturer’s instructions as previously described [18].

### 2.5. Determination of PKA Enzyme Activity

Nine microliters of crude protein extract from 24 h *A. fumigatus* cultures normalized by the Bradford method were added in triplicate to reaction mixtures containing fluorescent dye-coupled PepTag^®^ A1 Peptide LRRASLG (Promega, Madison, WI, USA) as a test substrate, and incubated for 30 min at 25°C. Activity was assessed per manufacturer’s instructions based on migration of negatively charged, phosphorylated peptide towards the anode of an agarose gel. Qualitative assessment was based on relative brightness of phosphopeptide bands observed under UV light. Quantitative comparison of relative PKA activity between samples was accomplished by analyzing the UV gel image using ImageJ (v. 2.0.0-rc-68/1.52e) open-source image-processing software. Cumulative brightness measurements of equal-sized band regions were collected and averaged between triplicate reactions for each experimental condition. Statistical comparisons were made using Student’s *t*-tests, and *p* values of <0.05 were considered significant.

### 2.6. Fluorescence Microscopy

Conidia (10^3^) of the respective RFP-labeled strains were cultured in 35 mm glass-bottomed petri dishes containing 3 mL of liquid GMM, and incubated for 16 h at 37 °C prior to visualization. For carbon and nitrogen starvation, GMM was removed via pipetting at 14 h, and replaced with 3 mL of carbon and nitrogen-free medium, followed by incubation for an additional 2 h. Hyphae were visualized using an Axio Observer 3 microscope (Carl Zeiss, Oberkochen, Germany) equipped with ZEN Lite imaging software.

### 2.7. Atg24 Interactome and Bioinformatic Analysis of Protein Functional Cluster Enrichment

GFP-Trap^®^ affinity purification was performed as described above for crude cellular extracts from triplicate cultures of an *A. fumigatus* strain expressing GFP-labeled Atg24 protein (Atg24-GFP), and a control strain expressing free GFP. Quantitative LC-MS/MS was performed on affinity-purified proteins, as described above. Following database searching and peptide scoring using Proteome Discoverer validation, the data were annotated at a 1% protein false discovery rate. Trim Mean normalization was used (SLTMM) to adjust the total signal in each sample to the mean of all signals (excluding the highest and lowest 10%). To assess interaction probability, peptide levels were compared between triplicates of Atg24-GFP and GFP control strains via *t*-test. Interactions were considered significant if *p* < 0.05, and interactant abundance was greater than 2-fold higher in Atg24-GFP samples vs. GFP control samples. The set of significant interactants was analyzed for enrichment of particular functional categories using the FungiFun v2.2.8 (https://elbe.hki-jena.de/fungifun/, accessed 10 February 2022) bioinformatic program, which compared sets of significantly upregulated or downregulated proteins to the complete set of annotated *A. fumigatus* (strain Af293) proteins to identify disproportionate enrichment of proteins associated with particular functions based on the FunCat categorization scheme. Enrichment was calculated using Fisher’s exact test with a significance level cutoff of *p* < 0.05.

### 2.8. Murine Invasive Aspergillosis Virulence Assays and Histopathological Analysis

Mice weighing 13–16 g (CD1, Charles River Laboratory, Raleigh, NC, USA) were immunosuppressed with cyclophosphamide (150 mg/kg intraperitoneally, days –2 before infection and +3 post-infection) and triamcinolone acetonide (40 mg/kg, subcutaneously, days −1 and +6). For infections, 40 μL of 1 × 10^8^ conidia/mL suspensions of the relevant strains were delivered intranasally following a brief isoflurane anesthesia induction. Survival was plotted on a Kaplan–Meier curve, and analyzed using log rank pair-wise comparison. *p*-values less than 0.05 were considered statistically significant. To characterize disease histopathology, additional mice were infected with each of the analyzed strains. Mice were euthanized on day +3 after infection, and lungs harvested. Lung sections were stained with Gomori’s methenamine silver stain to visualize fungal hyphae, and with hematoxylin and eosin stain to examine inflammation and tissue damage, as previously described [22]. Animal studies at Duke University Medical Center were in full compliance with all of the guidelines of the Duke University Medical Center Institutional Animal Care and Use Committee (IACUC), and in full compliance with the United States Animal Welfare Act (Public Law 98–198). Duke University Medical Center IACUC approved all of the vertebrate studies under the protocol number A-249-16-11. The studies were conducted in the Division of Laboratory Animal Resources (DLAR) facilities, which are accredited by the Association for Assessment and Accreditation of Laboratory Animal Care (AAALAC).

## 3. Results

### 3.1. Advanced Whole-Proteomic and Phosphoproteomic Approaches Identify Autophagy Proteins as Potential PKA-Dependent Effectors

In order to better understand the mechanisms of PKA regulation of autophagy, we examined our whole proteomic and phosphoproteomic LC-MS/MS data in greater detail, and compiled an index of specific PKA-regulated proteins with known or putative roles in autophagy. We included proteins in this index that showed statistically significant (*p* < 0.05) alteration of total protein abundance (regulation of expression) > 2-fold between WT and Δ*pkaC1* strains in the open whole proteome dataset, or alteration of phosphoprotein abundance (phosphoregulation) > 2-fold in the phosphopeptide-enriched dataset (Figure 1). Numerous autophagy-associated proteins were thus identified as PKA-regulated, either directly or indirectly via phosphorylation and/or modulation of expression level. This analysis revealed five proteins with likely functions in autophagy showing PKA-dependent alteration of protein levels (Table 1), and 12 autophagy-associated proteins with significantly altered phosphorylation in the WT vs. Δ*pkaC1* backgrounds, indicating them as targets of PKA phosphoregulation (Table 2), with 16 total unique proteins between the sets.

Strategy for discerning likely autophagy-associated PKA regulated proteins among proteins identified in LC-MS/MS analysis. The total set of identified proteins (from open protein dataset) and phosphoproteins (from both TiO_2_ and antibody-enriched datasets) were narrowed to those showing differing abundance of at least 2-fold in WT compared to Δ*pkaC1* genetic background, with a statistical significance level of *p* < 0.05. This set was further narrowed to those proteins with putative or known functions in autophagy, resulting in a set of 16 likely PKA-regulated autophagy proteins.

### 3.2. PKA Regulates Nutrient Sensing and Autophagosome Formation and Exhibits Reciprocal Regulation with the Key Autophagy Marker Protein Atg8

In *A. fumigatus,* protein and carbohydrate catabolic pathways are activated during invasive growth within the host nutrient-limited environment of the lung. Gene expression profiles under these conditions mimic those generated by carbon and nitrogen starvation, two key conditions inducing autophagy. To first elucidate the primary role of PKA in regulating autophagy in *A. fumigatus*, we examined the impact of PKA activity on autophagosome formation by expressing an RFP-labeled conserved late-autophagosome marker component Atg8 using the native *atg8* promoter at the native locus in our Δ*pkaC1* mutant background. Deletion of Atg8 resulted in reduced conidiation on glucose minimal medium agar (GMM), but no reduction in radial growth rate (Figure 2A). We found that under basal growth conditions, Atg8 expressed in the WT background remained diffuse in the cytosol, but under carbon and nitrogen starvation (C/N-) conditions, Atg8 localized to punctate autophagosomal structures (Figure 2B). In contrast, loss of PKA activity resulted in constitutive autophagosome formation and punctate Atg8 localization under both basal and C/N- conditions (Figure 2B), indicating dysregulation of autophagy in the absence of PKA. These findings also correlated with the expression of Atg8, as determined by western blotting using an anti-RFP antibody. In the WT background, Atg8 was expressed at a low level under basal conditions, and expression increased during C/N starvation, whereas in the Δ*pkaC1* background, Atg8 was constitutively expressed at a high level regardless of C/N abundance (Figure 2C). Reciprocally, loss of Atg8 protein also increased overall PKA activity in cell extracts, as determined via in vitro fluorescent substrate phosphorylation and quantitative analysis of gel band intensity, an effect which was further enhanced under autophagy-inducing nutrient starvation conditions (Figure 2D). These key findings indicate PKA control over core autophagy processes during *A. fumigatus* growth under nutrient-limited conditions relevant to pathogenesis.

### 3.3. Atg20 Interacts with Atg24 to Control A. fumigatus Growth and Pathogenesis

In our previous proteomic analysis, we identified the putative autophagy-associated sorting nexin protein Atg24 as a direct phosphorylation target of PKA [11]. Atg24 was also demonstrated to be important for the virulence of *A. fumigatus* in a murine infection model, and PKA-dependent phosphorylation at specific amino acid residues was shown to regulate the fungal cell wall stress response. In order to better understand the cellular function of this key protein, we interrogated the Atg24 interactome by GFP-Trap affinity purification of GFP-labeled Atg24 from whole cell extracts, followed by LC-MS/MS analysis to quantitatively identify co-purified interacting proteins. Using this methodology, we identified 203 Atg24 interacting proteins (Appendix A). Functional category annotation of these interactants using the FungiFun analytical software revealed enrichment of proteins involved in: protein targeting, sorting, and translocation; lipid binding; protein/peptide degradation; peptide binding; nitrogen, sulfur, and selenium metabolism; protein modification; mitochondrial function; and hyphal tip growth (Figure 3A). Atg8 was not identified as interacting with Atg24 in this analysis. However, among the set of identified Atg24 interactants, another putative autophagy protein, Atg20, was indicated as the primary interactant of Atg24, based on the high number of co-purified peptides for this protein (Appendix A). This interaction may be direct or indirect, although the presence of BAR dimerization domains in both proteins suggests direct interaction is likely possible. Strikingly, we also identified Atg20 as a likely phosphorylation target of PKA in our whole phosphoproteomic analysis (Table 2). Atg20 is predicted to also be a sorting nexin protein with a domain structure similar to that of Atg24, including PX phosphoinositide (membrane)-binding and BAR dimerization domains (Figure 3B). Additionally, the Atg20 primary amino acid sequence includes four canonical PKA target motifs (R/K-R/K-X-S/T), two of which, serine 132 (S132) and serine 525 (S525), we identified as demonstrating PKA-dependent phosphorylation in our phosphoproteomic analysis. Deletion of the *atg20* gene resulted in a strong reduction in radial growth rate very similar to that previously observed for *atg24* deletion on GMM agar medium (Figure 3C). Simultaneous deletion of both *atg20* and *atg24* produced a similar growth defect to either of the individual gene deletions, suggesting that the functionality of each protein is interdependent on the presence of the other. Consistent with this hypothesis, the Δ*atg20* strain also showed a loss of paradoxical growth recovery normally observed in the presence of a high concentration (4 µg/mL) of the cell-wall-targeting antifungal caspofungin (CSP) in comparison to a lower concentration (0.5 µg/mL), similar to that of Δ*atg24*, a phenotype which was not altered in the Δ*atg20*; Δ*atg24* double deletion mutant (Figure 3C). All strains showed similar sensitivity to an optimally fungistatic concentration of CSP (0.5 µg/mL). Given the predicted roles of these proteins in autophagy, we also examined growth under autophagy-inducing nitrogen starvation conditions, but growth inhibition from deletion of either *atg20*, *atg24*, or both together, was found to be merely proportional to that under basal growth conditions, therefore not indicating an enhanced role for these proteins in growth during nitrogen limitation (Figure 3D). Stronger growth defects observed in Atg20 and Atg24 deletion strains compared to the Atg8 deletion strain support roles for the former proteins in functions beyond core autophagy processes. In contrast to our findings for Atg24, wherein mutation of PKA target sites resulted in an altered response to cell wall stress from CSP exposure, here we found that mutation of S525 of Atg20 to a nonphosphorylatable alanine residue did not alter sensitivity of the fungus to 0.5 µg/mL CSP, or impair paradoxical growth at 4 µg/mL CSP (Figure 3D). This suggests that PKA-dependent phosphorylation at S525 may not be functionally relevant to cell wall stress tolerance, or that it may require synergistic phosphorylation at additional sites to exert its effects, as has previously been observed for phosphoregulation of other *A. fumigatus* proteins, including the transcription factor CrzA [23].

In order to assess the contribution of Atg20 to *A. fumigatus* pathogenesis, we employed a murine model of IA. Immunosuppressed mice were infected with spores of WT, Δ*atg20*, and an Atg20 complementation strain (Atg20-Comp) via intranasal inoculation, and monitored for survival for 14 days. Mortality of mice infected with Δ*atg20* (50%) was found to be significantly lower than that observed either for the WT (90%) or Atg20-Comp strain (100%) (Figure 4A). This reduction in virulence is comparable to that previously observed for Atg24 deletion, further supporting interdependent functions for the two proteins. Histological analysis of lung tissue from mice infected with each of the three strains revealed reduced invasive hyphal growth and tissue damage in mice infected with Δ*atg20*, in contrast to more extensive invasion and damage in tissues infected with the WT and Atg20-Comp strains (Figure 4B). Together, these findings indicate an important role for Atg20 in pathogenesis in a mammalian host system.

### 3.4. PKA Regulates Key Cellular Recycling Pathway Proteins Contributing to Fungal Growth

In addition to Atg20, numerous other uncharacterized and potentially important autophagy-associated proteins were also identified as probable direct or indirect PKA targets with PKA-dependent phosphorylation or regulation of gene expression. In order to efficiently characterize proteins within this set with the greatest potential impact on *A. fumigatus* growth and pathogenesis, and potential clinical relevance, we prioritized proteins for study based on several factors, including fungal-specificity (low sequence similarity to any human proteins), potential role in autophagy based on evidence in *A. fumigatus* or other fungi, degree to which phosphorylation level or protein expression level is impacted by PKA, and confirmed phosphorylation of canonical PKA target motif residue(s), indicating a likely direct PKA target. The eight highest priority proteins determined using this strategy are shown in Table 3. Deletion strains were generated for each of these eight proteins in order to assess their impact on *A. fumigatus* growth and development. Though the majority of these deletions did not significantly impact growth, three did produce strong reductions in radial growth rate on GMM agar (Figure 5). These included the putative vacuolar sorting protein Vps17 (0% human similarity), a putative ubiquitin C-terminal hydrolase (RefSeq ID XP_755178.1, 10% human similarity), both displaying PKA-dependent phosphorylation (albeit not at canonical PKA target motifs, suggesting indirect rather than direct phosphoregulation), and a putative vacuolar H^+^/Ca^2+^ exchanger (RefSeq ID XP_755098.1, 0% human similarity) regulated by PKA at the expression level (Table 3). Growth inhibition of these deletion strains on nitrogen-free media was found to be proportional to that under basal growth conditions, thus as above for Atg20 not indicating an enhanced role for these proteins under nitrogen limitation conditions (Figure 5). Further characterization of PKA regulation of these identified autophagy proteins is required to understand how these effectors function in *A. fumigatus* growth and pathogenesis.

## 4. Discussion

Fungal pathogens encounter nutrient-limited conditions within the host, resulting in upregulation of protein and carbohydrate catabolic pathways during invasive growth for efficient utilization of available nutrients [13,24]. Our phosphoproteome data suggest PKA controls these key processes via phosphorylation–dephosphorylation and regulating expression levels of effectors. In this study, we have characterized novel aspects of the regulation of autophagy-associated proteins by PKA in *A. fumigatus* (Figure 6). Our data indicate that PKA regulates the autophagy pathway much more extensively than previously known, including targeting of novel effector proteins with fungal-specific functions important for invasive disease.

The enhancement of autophagosome-like formation as evidenced by localization of the conserved autophagosomal marker protein Atg8, as well as increased expression of Atg8 in the absence of PKA (Δ*pkaC1* strain), suggests a general role for PKA in inhibition of autophagy during growth in nutrient rich environments, consistent with observations in yeast wherein PKA has been found to negatively phosphoregulate the key autophagy protein Atg1 [14,15]. However, it is important to more clearly define the role of PKA in regulating autophagy activity in future studies via more direct methods, such as a proteolytic cleavage-assay utilizing RFP-Atg8 [25]. In the present study, we further define multiple additional autophagy-associated effectors not previously identified as regulated by PKA in any species. We also demonstrate a role for PKA in regulation of autophagy proteins at the gene expression level in addition to the phosphoregulatory level, both through our observation of PKA-dependent suppression of Atg8 expression during growth under nutrient-rich conditions, and through our whole proteomic analysis of PKA-dependent alterations in protein abundance. Furthermore, as illustrated in our model (Figure 6), autophagy-inducing nutrient starvation conditions stimulate PKA activity, whereas autophagy proteins themselves, such as Atg8, both inhibit and are inhibited by PKA in a reciprocal manner, as evidenced by enhanced enzymatic activity of PKA in the absence of Atg8 under both basal and nutrient-limited conditions. The nutrient-mediated and Atg8-mediated mechanisms of PKA regulation appear to operate at least partially independently given their observed additive effects on PKA activity. This increased PKA activity observed in response to nutrient limitation could potentially indicate a mechanism wherein PKA serves as a check on autophagy, working antagonistically with activating factors under these conditions. Further analysis is required to elucidate the specific mechanisms involved in these processes of autophagy activation and repression.

In analyzing the interactome of Atg24, which we previously characterized as a PKA phosphorylation target important for *A. fumigatus* virulence, we identified 203 significant protein interactors (Appendix A) which were enriched in several functional categories (Figure 3A). The most highly enriched category was that of protein sorting and targeting, consistent with the identification of Atg24 as a putative sorting nexin. Strong enrichment for the functional category of protein degradation also supports a role for Atg24 in autophagy. It is also striking that mitochondrial function was another highly enriched category, suggesting that Atg24 may play an important role in mitophagy, as described previously for the homologous protein of the plant pathogenic fungus *Magnaporthe oryzae* [26]. Furthermore, our interactome analysis led us to describe, for the first time, the putative autophagy-associated sorting nexin Atg20, and to demonstrate an important role for this novel protein in *A. fumigatus* growth, pathogenesis, and cell wall stress response. We have demonstrated (either direct or indirect) physical and functional interaction between Atg20 and Atg24, both of which we have also shown to undergo PKA-dependent phosphorylation. However, the cellular functions of these proteins leading to the observed phenotypic outcomes remain to be discovered. The nearest *S. cerevisiae* homolog of *A. fumigatus* Atg20 is the sorting nexin SNX41p, which has been demonstrated to form a complex with sorting nexins SNX42p and SNX4p, the latter of which is the nearest yeast homolog of *A. fumigatus* Atg24, suggesting a potentially parallel role for these respective proteins in the two species. In yeast, this complex functions in the retrieval of proteins from post-golgi endosomes prior to degradation in the vacuole [27]. In *A. fumigatus*, it is therefore possible that the Atg20/Atg24 complex may similarly function in protein retrieval rather than in facilitating autophagic degradation. More work is needed to determine whether this is the case and how this functionality contributes to fungal growth and, in particular, invasive growth within the mammalian host lung environment leading to disease. Examination of core autophagy processes, such as autophagosome formation using RFP-Atg8 in Atg20 and Atg24 deletion backgrounds, would also be instructive. Additionally, it is of interest to better understand the molecular mechanisms of Atg20/Atg24 regulation of the fungal response to cell wall stress, and how PKA phosphoregulation of both proteins influences this response. Such understanding could potentially lead to the development of enhanced cell-wall-targeting treatment strategies for IA.

In this work, we report preliminary characterization of several additional PKA-regulated autophagy-associated proteins with important contributions to hyphal growth in *A. fumigatus*. The putative sorting nexin Vps17 exhibits homology to a member of the retromer complex of *S. cerevisiae*, involved in protein retrieval from endosomes [28]. Intriguingly, this similar functionality to yeast homologs of Atg20 and Atg24 suggests a possible broader role for PKA in regulating the endosomal retrieval pathway of *A. fumigatus*. Further characterization of this potential role, as well as the functional importance of possibly indirect PKA phosphoregulation of Vps17, is required. Less evidence is available concerning the potential function of the putative ubiquitin C-terminal hydrolase. This protein displays only minimal similarity to characterized proteins in other fungal species, including the closest yeast homolog Ubp15p, which functions in peroxisomal import [29]. However, due to the low similarity between these proteins, functional comparisons may be unreliable. Finally, the putative vacuolar H^+^/Ca^2+^ exchanger displays moderate similarity to yeast Vcx1p, which functions in vacuolar Ca^2+^ accumulation, and is negatively regulated by calcineurin through a mechanism independent of the transcription factor Crz1p [30,31,32]. It will be of interest to further examine the function of this protein in *A. fumigatus*, as well as its regulation by PKA. Deciphering the mechanism of PKA regulation of growth and virulence via novel effectors associated with autophagy pathways will facilitate identification of fungal-specific PKA targets and novel pathogenic mechanisms, and offer future therapeutic translational potential. This work should serve as a basis for future in-depth studies elucidating the role of cellular recycling pathways in fungal physiology and pathogenesis.

## Figures and Tables

**Figure 1 jof-08-00354-f001:**
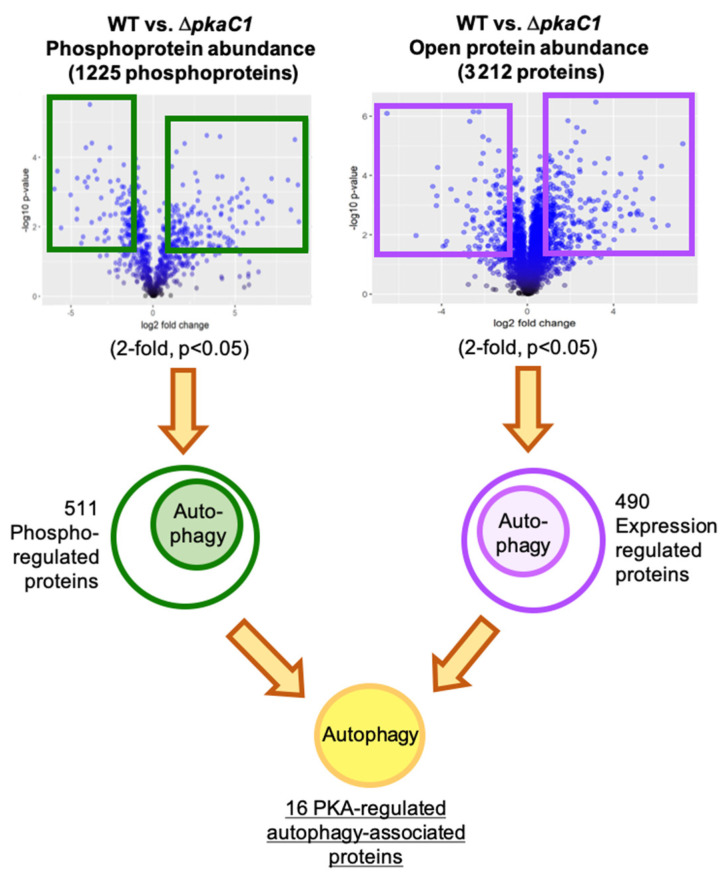
Identification of PKA-regulated autophagy proteins.

**Figure 2 jof-08-00354-f002:**
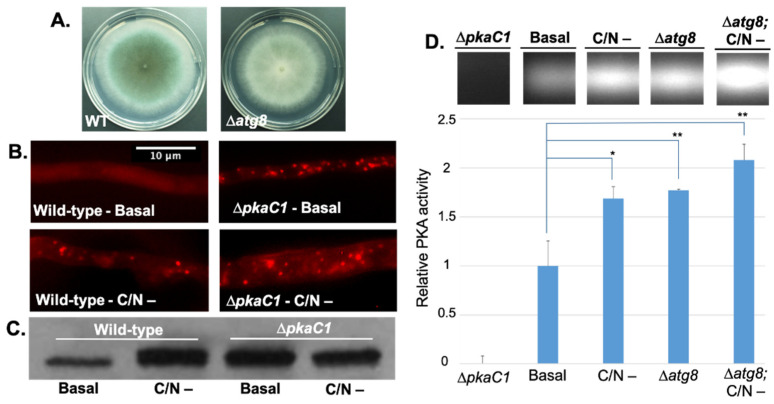
Broad interaction between PKA and autophagy pathway. (**A**) Reduced conidiation phenotype of Atg8 deletion strain (Δ*atg8*) grown on GMM agar for 4 days at 37°C, compared to the wild-type (WT) strain. (**B**) Localization of RFP-labeled Atg8 protein expressed in WT and Δ*pkaC1* genetic backgrounds under basal and carbon/nitrogen starvation conditions (C/N-). Strains were cultured in glass-bottomed petri dishes containing liquid GMM, and incubated for 16 h at 37 °C prior to visualization. For carbon and nitrogen starvation, GMM was removed via pipetting at 14 h, and replaced with carbon and nitrogen-free medium, followed by incubation for an additional 2 h. Punctate structures indicate the presence of autophagosomes. (**C**) Abundance of RFP-labeled Atg8 protein expressed in WT and Δ*pkaC1* genetic backgrounds under basal and carbon/nitrogen starvation conditions (C/N-). Proteins from normalized crude extracts were separated via SDS-PAGE, and hybridized to a PVDF membrane, and then probed with anti-RFP primary antibodies to detect Atg8 protein in a western blot assay. Atg8 expression was upregulated by either C/N starvation or loss of PKA catalytic subunit. (**D**) PKA activity in WT, Δ*atg8*, and Δ*pkaC1* (negative control) genetic backgrounds under basal and C/N starvation conditions. Protein-normalized crude cellular extracts were added in triplicate to reaction mixtures containing fluorescent dye-coupled PepTag peptide (Promega) as a PKA-specific test substrate, and incubated for 30 min at room temperature. Activity was assessed based on migration of negatively charged, phosphorylated peptide towards the anode of an agarose gel. PKA activity was cumulatively enhanced by both C/N starvation and loss of Atg8 protein. Gel band brightness was quantified using ImageJ image analysis software, and comparisons between averaged triplicates for each condition were made using Student’s *t*-tests (* = *p* < 0.05, ** = *p* < 0.01, error bars represent standard deviations).

**Figure 3 jof-08-00354-f003:**
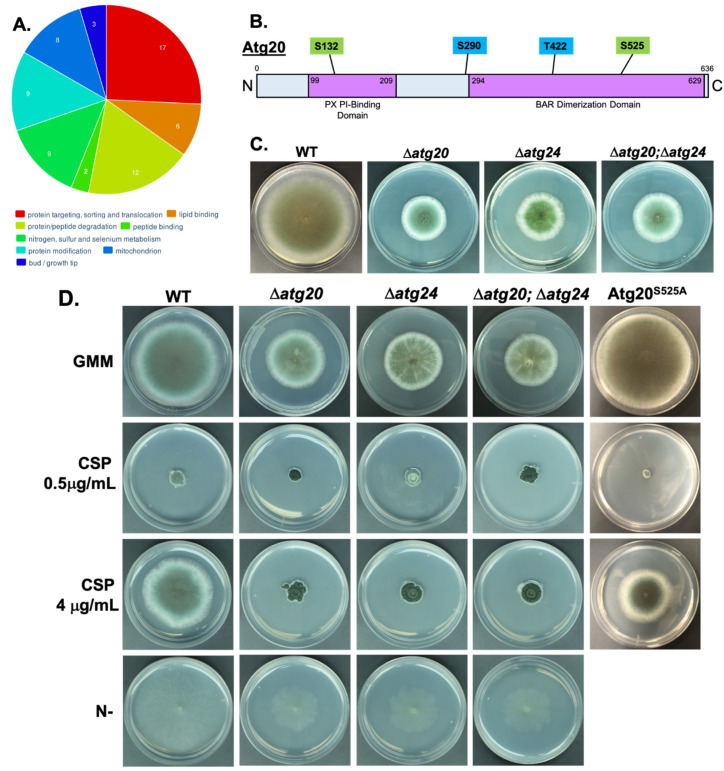
Atg24 interactome, Atg20 structure, deletion, and cell wall stress phenotypes. (**A**) Pie chart of enriched functional categories for Atg24-interacting proteins identified through GFP-Trap affinity purification, and LC-MS/MS analysis demonstrating at least 2-fold greater abundance in the Atg24-GFP genetic background compared to GFP control background, and statistical significance of *p* < 0.05 in *t*-test comparison. The set of interactant proteins was compared to the total set of annotated *A. fumigatus* proteins based on the FunCat classification systems, as analyzed using the FungiFun 2.2.8 bioinformatic software. Numbers on chart represent numbers of proteins associated with each category. Interactant proteins not associated with enriched functional categories are not represented. (**B**) Diagram of Atg20 primary amino acid sequence indicating putative functional domains, locations of canonical PKA target motif serine (S) and threonine (T) residues, and identified sites of phosphorylation (green boxes). Blue boxes indicate PKA target motifs not identified as phosphorylated. (**C**) Reduced radial growth phenotype of Atg20 deletion strain (Δ*atg20*), Atg24 deletion strain (Δ*atg24*), and Atg20/Atg24 double deletion strain (Δ*atg20*; Δ*atg24*) grown on GMM agar for 4 days at 37°C, compared to the wild-type (WT) strain. (**D**) Comparison of radial growth of WT, Δ*atg20,* Δ*atg24*, and Δ*atg20*; Δ*atg24* deletion strains, as well as Atg20 serine 525 to alanine (Atg20^S525A^) mutant strain on GMM agar infused with 0, 0.5, or 4 µg/mL caspofungin (CSP), and nitrogen free GMM for indicated strains, for 5 days at 37 °C.

**Figure 4 jof-08-00354-f004:**
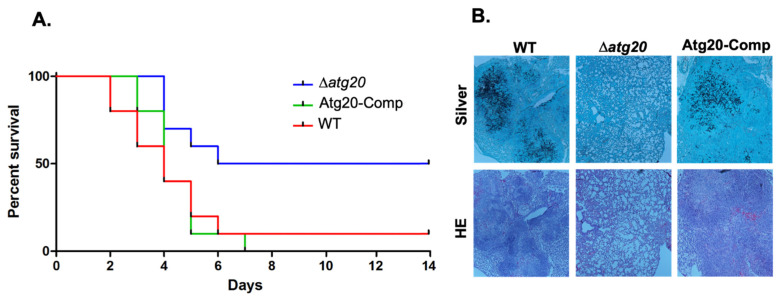
Atg20 deletion strain virulence in murine invasive aspergillosis model. (**A**) Survival curve for Atg20 mutant strains in intranasal murine model of invasive pulmonary aspergillosis. Ten mice immunosuppressed with 150 mg/kg cyclophosphamide; 40 mg/kg triamcinolone were inoculated with 40 µL of 10^8^ spores/mL suspensions of WT (Ku80), Δ*atg20*, or Atg20-Comp strain conidia. Mice were monitored for survival for 14 days following infection. Mortality was significantly reduced for mice infected with Δ*atg20* compared to WT (*p* = 0.0258) and Atg20-Comp (*p* = 0.0091) strains, as determined by log-rank comparison. (**B**) Histological analysis of HE and silver stained mouse lung tissue 3 days after infection with WT, Δ*atg20*, or Atg20-Comp conidia from experiment shown in panel **A**. Dark areas in silver stain indicate hyphal growth; dark purple and red areas in HE stain indicate tissue damage.

**Figure 5 jof-08-00354-f005:**
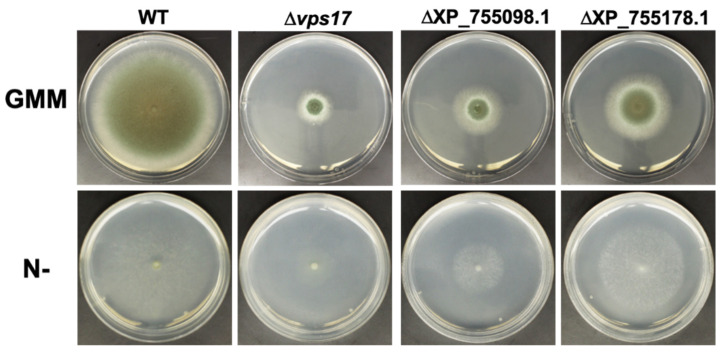
Deletion phenotypes of additional PKA-regulated autophagy proteins. Reduced radial growth phenotypes of Vps17 (Δ*vps17*), putative vacuolar H^+^/Ca^2+^ exchanger (ΔXP_755098.1), and putative ubiquitin C-terminal hydrolase (ΔXP_755178.1) deletion strains grown on GMM agar or nitrogen-free GMM agar (N-) for 4 days at 37 °C, compared to the wild-type (WT) strain.

**Figure 6 jof-08-00354-f006:**
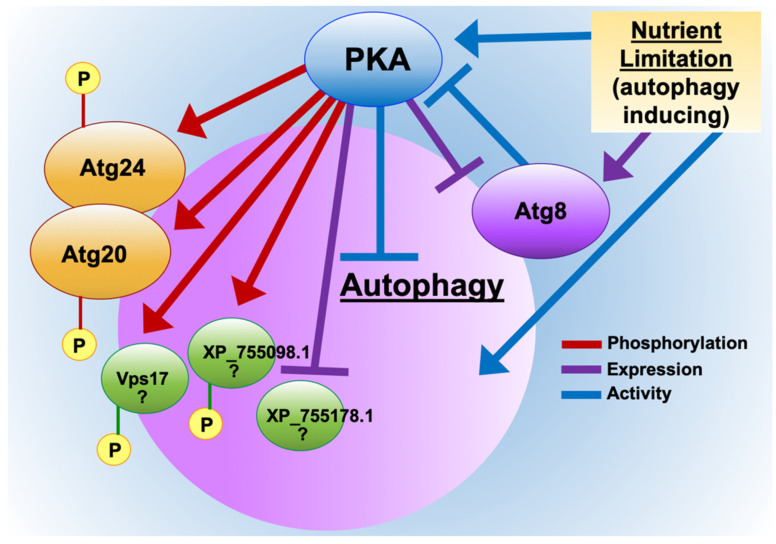
Schematic model showing PKA regulation of autophagy in *A. fumigatus*. Identified autophagy-associated proteins, processes, and conditions regulating and regulated by PKA with regard to phosphorylation, protein expression, and activity, as indicated. “P” indicates phosphorylation of the effector proteins by PKA. Arrows represent positive regulation, whereas bars represent negative regulation. Question marks indicate putative nature of additional proteins’ involvement in autophagy. Regulation of phosphorylation, expression, or activity may be direct or indirect in each case depicted.

**Table 1 jof-08-00354-t001:** Likely autophagy-associated proteins exhibiting PKA-dependent expression.

RefSeq ID	Protein Name	Description	WT/Δ*pkaC1* Fold Change	*p*-Value
XP_755360.1	TorA	TOR pathway phosphatidylinositol 3-kinase	15.3	0.025
XP_753398.1	NA	N-acetylglucosaminyl-phosphatidylinositol deacetylase	6.68	0.026
XP_752894.1	Atg27	autophagy protein	−2.33	0.0395
XP_751230.1	CpyA/Prc1	carboxypeptidase	−2.79	0.002
XP_755098.1	NA	vacuolar H^+^/Ca^2+^ exchanger	−9.47	0.049

NA = Not available.

**Table 2 jof-08-00354-t002:** Likely autophagy-associated proteins exhibiting PKA-dependent phosphorylation.

RefSeq ID	Protein Name	Master Protein Descriptions	WT/Δ*pkaC1* Fold Change	*p*-Value
XP_748106.1	VpsA	vacuolar dynamin-like GTPase	263.31	0.0003
XP_753919.2	Vps17	vacuolar protein sorting-associated protein	63.29	0.0093
XP_751536.1	Atg24	vacuolar targeting sorting nexin protein	25.3	0.0020
XP_001481667.1	NA	RING finger protein	12.78	0.0337
XP_755360.1	TorA	TOR pathway phosphatidylinositol 3-kinase	7.42	0.0156
XP_751003.1	Vps15	vacuolar protein kinase	6.69	0.0126
XP_749349.1	Cue3	CUE domain protein	4.1	0.0349
XP_755749.2	Atg20	autophagy sorting nexin protein	3.58	0.0008
XP_750137.1	Scn1	Cut9 interacting protein Scn1	3.04	0.0132
XP_755178.1	NA	ubiquitin C-terminal hydrolase	2.29	0.0332
XP_753300.1	Vac8	vacuolar armadillo repeat protein	−2	0.0200
XP_755238.1	Vtc4	vacuolar transporter chaperone	−6.67	0.0045

NA = Not available.

**Table 3 jof-08-00354-t003:** Prioritized PKA-regulated autophagy-associated proteins.

RefSeq ID	Protein Name	Description	Human Similarity	Likely Autophagy Relevance	PKA Motif Phos.	Phos. Fold Change	Expr. Fold Change	*p*-Value
XP_753919.2	Vps17	Vacuolar protein sorting-associated protein	0%	High	No	63.29	-	0.0093
XP_755098.1	NA	Vacuolar H^+^/Ca^2+^ exchanger	0%	High	No	-	−9.47	0.049
XP_001481667.1	NA	RING finger protein	5%	Moderate	Yes	12.78	-	0.0337
XP_755178.1	NA	Ubiquitin C-terminal hydrolase	10%	High	No	2.29	-	0.0332
XP_755238.1	Vtc4	Vacuolar transporter chaperone	7%	High	No	−6.52	-	0.0045
XP_750137.1	Scn1	Cut9 interacting protein	7%	Moderate	Yes	3.04	-	0.0132
XP_749349.1	Cue3	CUE domain protein	15%	Moderate	Yes	4.1	-	0.0349
XP_752894.1	Atg27	Autophagy protein	0%	High	No	-	−2.33	0.0395

NA = Not available.

## Data Availability

Raw proteomic data are accessible at the Duke Center for Genomic and Computational Biology Express Data Repository. PKA phosphoproteome data: https://discovery.genome.duke.edu/express/resources/5081/Atg24 (accessed on 10 February 2022), interactome data: https://discovery.genome.duke.edu/express/resources/5492/ (accessed on 10 February 2022).

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
