# Peer review of "Protein Kinase A Regulates Autophagy-Associated Proteins Impacting Growth and Virulence of Aspergillus fumigatus"

_jof, 2022, doi:10.3390/jof8040354_

Round 1

Reviewer 1 Report

Shwab et al. have identified Atg20 as a potential phosphorylation target of protein kinase A (PKA) in Aspergillus fumigatus. They have shown Atg20 is important for virulence; however, direct evidence of Atg20 phosphorylation by PKA and functional relevance of Atg20 phosphorylation were missing.

L229-230: The information of how the strain expressing RFP-labelled Atg8 was made is missing. Was the expression of RFP-Atg8 regulated under the native promoter?

L274: Why is the total number in Fig. 3A not 203?

L284: Fig. 3B

Fig. 3B: What do blue boxes indicate?

L287-290: Show the alignment result.

L292: Fig. 3C

Fig. 3C: Please describe why growth phenotypes were different between Atg8 and Atg20 and Atg24.

L298, 303: Fig. 3D

Fig. 6: No direct physical interaction data between Atg20 and 24 are shown. The interaction might be indirect. No direct data regarding phosphorylation on Atg20 by PKA are shown. This phosphorylation might be indirect. The information on how 3 green proteins are involved in autophagy is totally missing. This figure indicates PKA negatively regulates autophagy; however, authors just showed autophagosome-like formation was enhanced in delta pkaC1. RFP-Atg8 cleavage assay is needed to clearly investigate autophagy activity.

L437-439: Analyzing RFP-Atg8 in delta atg20 or atg24 can tell us more information.

Reviewer 2 Report

This is an interesting study of characterizing protein kinase A regulated autophagy-associated proteins in Aspergillus fumigatus, an airborne opportunistic fungal pathogen, and associated virulence. The authors have identified a novel autophagy protein, Atg20, and studied its role in virulence using in vivo mouse model.

Comments:

  1. Line 123: expand CSP, caspofungin acetate, as it appears for the first time in the text.
  2. Lines 151/section 3.2: it is mentioned about GFP-labeling of proteins in the method section; however, RFP is mentioned here. Whether both GFP and RFP-labeled strains were generated?
  3. Figure 1: It will be nice if the authors indicate the numbers of proteins identified in each category, particularly in level 2 showing phopho-regulated and expression regulated proteins.
  4. Figure 3D: It is surprising to see that the WT strain showed radial growth inhibition at 0.5 mg/mL but at 4 mg/mL of CSP.
  5. Is there any interaction between Atg8 and Atg20/Atg24?
